# An empirical model to calculate snow depth from daily snow water equivalent: SWE2HS 1.0

Johannes Aschauer[1], Adrien Michel[1,2], Tobias Jonas[1], and Christoph Marty[1]

[1]WSL Institute for Snow and Avalanche Research SLF, Davos, Switzerland

[2]School of Architecture, Civil and Environmental Engineering, École Polytechnique Fédérale de Lausanne (EPFL), Lausanne, Switzerland

**Correspondence:** Johannes Aschauer (johannes.aschauer@slf.ch)

**Abstract.** Many methods exist to model snow densification in order to calculate the depth of a single snow layer or the depth of the total snow cover from its mass. Most of these densification models need to be tightly integrated with an accumulation and melt model and need many forcing variables at high temporal resolution. However, when trying to model snow depth on climatological timescales, which is often needed for winter tourism related applications, these preconditions can cause barriers. Often, for these types of applications empirical snow models are used. These can estimate snow accumulation and melt based on daily precipitation and temperature data, only. To convert the resultant snow water equivalent time series into snow depth, we developed the empirical model SWE2HS. SWE2HS is a multi-layer densification model which uses daily snow water equivalent as sole input. A constant new snow density is assumed and densification is calculated via exponential settling functions. The maximum snow density of a single layer changes over time due to overburden and SWE losses. SWE2HS has been calibrated on a data set derived from a network of manual snow stations in Switzerland. It has been validated against independent data derived from automatic weather stations in the European Alps (Austria, France, Germany, Switzerland) and against withheld data from the Swiss manual observer station data set which was not used for calibration. The model fits the calibration data with root mean squared error (RMSE) of 8.4 cm, coefficient of determination ($R^2$) of 0.97, and BIAS of -0.3 cm and is able to achieve RMSE of 20.5 cm, $R^2$ of 0.92, and BIAS of 2.5 cm on the automatic weather stations validation data set and RMSE of 7.9 cm, $R^2$ of 0.97, and BIAS of -0.3 cm on the manual stations validation data set. The temporal evolution of the bulk density can be reproduced reasonably well on all three data sets. Due to its simplicity, the model can be used as post-processing tool for output of any other snow model that provides daily snow water equivalent output. Owing to its empirical nature, SWE2HS should be only used in regions with a similar snow climatology as the European Alps or has to be recalibrated for other snow climatological conditions. SWE2HS is available as a Python package which can be easily installed and used.

## 1 Introduction

Seasonal snow cover is an important variable with regard to ecology, water resource management, and tourism industry. Accordingly, a large range of models of different complexity exist to calculate various properties of the snow cover. Traditionally, snow models were emerging from the hydrological community in order to estimate water resources from snow. Therefore,

the focus was set on snow water equivalent (SWE) of the snow cover for the first simple approaches such as the empirical temperature-index models in which the amount of melt is estimated by the sum of positive air temperatures (see e.g. Hock, 2003). Over time, more complex models were developed which are capable to calculate snow density, snow temperature profiles (Jordan, 1991), and snow microstructure (Lehning et al., 2002; Vionnet et al., 2012). Most of these more complex, physically based models require a rich set of input parameters such as incoming short and long wave radiation, wind speed, precipitation, and temperature at sub-daily temporal resolution. However, when applying models on longer timescales e.g. for climatological analyses, the required input parameters are often limited with regard to availability and temporal resolution. Accordingly, simpler empirical models are still often used for climatological analyses instead of employing more complex physically based models. Empirical models usually do not calculate snow depth (HS) and focus on SWE which is appropriate for climatological research on changes in hydrological regimes or glacier mass balance. However, when model output is addressed to stakeholders who are usually dealing with snow depth rather than snow water equivalent, such as in the traffic and the winter tourism sector, calculation of snow depth would be desirable. This applies mostly to spatially distributed model output, because due to the ease of measuring snow depth, point data are often available for specific sites.

Snow depth is the result of SWE and the bulk snow density ($\rho$), where $\mathrm{SWE} = \mathrm{HS} \cdot \rho$. Snow depth can be measured either manually by reading from snow stakes or automatically with lasers or ultrasonic devices (Kinar and Pomeroy, 2015). While modeling SWE requires the representation of snow mass accumulation and ablation, modeling snow depth needs to address different types of densification processes. These processes involve densification due to stress induced by overlying snow and metamorphic processes that change the size and shape of the snow crystals and thus affect snow density (Anderson, 1976). Metamorphic processes can be either destructive (at constant temperature), constructive (within a temperature gradient) or melt metamorphic (for melt refreeze cycles) (Sommerfeld and LaChapelle, 1970).

All densification models need to initialize the density of a snow layer or of the whole snowpack. Since there is yet no simple method to derive new snow density from a physical snowfall model, in snow models new snow density is either parameterized or kept at a fixed value. Various parameterizations exist and are usually based on estimating new snow density as a function of wind speed, temperature and relative humidity (see e.g. Helfricht et al., 2018; Valt et al., 2018). When applied on a daily resolution, the quality of such parameterizations is declining due to unknown timing of a snowfall event during the day and simultaneous occurrence of settling over the course of the day (Meister, 1986).

There exist several methods to model snow densification either per layer or for the entire snowpack which can be roughly classified into three categories. The first category is purely empirical whereby densification dynamics are described via exponential settling functions. This approach has first been proposed by Martinec (1956) and Martinec (1977) while variations of the method exist (e.g. Dawson et al., 2017; Koch et al., 2019; Essery, 2015; Aili et al., 2019; Brown et al., 2003, 2006). Dawson et al. (2017) for example use a non constant e-folding time of the settling rate based on air temperature with an additive overburden term, Essery (2015) use two different maximum densities for cold and melting snow where the exponential function converges to and Brown et al. (2006) use a maximum density based on snow depth. The second category of snow densification models is the semi-empirical method of Anderson (1976) which employs a two stage compaction due to metamorphism and pressure from overlying snow. The compaction due to stress uses a parameterized viscosity coefficient based on temperature.

Settling is enhanced when wet snow in the snowpack occurs. The scheme of Anderson (1976) has been adopted widely and is used in many snow and land surface models such as SNTHERM (Jordan, 1991), AMUNDSEN (Marke et al., 2015; Hanzer et al., 2016; Marke et al., 2018; Warscher et al., 2021), SNOWGRID-CL (Olefs et al., 2020) and CLM5 (van Kampenhout et al., 2017; Lawrence et al., 2019). Due to its need to determine wet snow in the snowpack, the method of Anderson (1976) has to be tightly integrated with a snow melt model. The third and most sophisticated category of snow densification models is using a snow viscosity coefficient which is parameterized based on temperature and/or microstructure of the snow. Snow compaction is then modeled by applying stress due to weight of overlying snow. This requires a complex physical model in order to be able to represent the processes which affect e.g. snow microstructure and is realized by the two physical energy balance models models Crocus (Brun et al., 1992; Vionnet et al., 2012) and SNOWPACK (Bartelt and Lehning, 2002; Lehning et al., 2002).

To our knowledge, none of the above described densification models can be easily used as a standalone model to transfer daily SWE to snow depth independently of the snow model, while many approaches exist to do the opposite, convert HS into SWE (e.g. Jonas et al., 2009; Winkler et al., 2021; McCreight and Small, 2014; Mizukami and Perica, 2008; Guyennon et al., 2019; Pistocchi, 2016). With new methods being developed to derive SWE from global navigation satellite system (GNSS) signal attenuation (Koch et al., 2019) or by cosmic ray attenuation (Gugerli et al., 2019), it would be even more desirable to be able to model snow depth from the derived SWE data (Capelli et al., 2022). For climatological applications, quantile mapping can be used to statistically correct the SWE output of simple melt and accumulation models that can be run far back in time using more complex models that require more input data and are therefore not suitable for climatological timescales up to several decades (Michel et al., 2023). To transfer the statistically corrected SWE field to HS, a standalone densification post-processing model would also be required. For these reasons, we developed a simple empirical snow densification model which uses daily SWE as sole forcing and transforms SWE to HS using exponential settling equations for individual layers inspired by Martinec (1977); Dawson et al. (2017); Koch et al. (2019); Essery (2015). We make an implementation of the model available as a Python package which can be downloaded and installed from the Python packaging index (PyPI).

The remainder of the paper is structured as follows. In Sect. 2 we describe the model as well as the technical implementation. In Sect. 3 we describe the data used for calibration and validation of the model alongside the used calibration methods. In Sect. 4 we show the performance of the calibrated model in alpine snow environments and discuss the scope and limitations of the model in Sect. 5.

## 2 Density model

### 2.1 Basic concept

The density model SWE2HS calculates snow depth at a daily resolution and is driven by the daily snow water equivalent of the snow cover only. In the following, we use the unit $\mathrm{mm\,w.e.}$ for SWE. The model creates a new layer with a fixed new snow density $\rho_{new}$ for every increase in SWE such that, over time, a snowpack of individual layers builds up. The density of a layer increases with an exponential decay function towards a time-varying maximum density. The maximum density starts with an

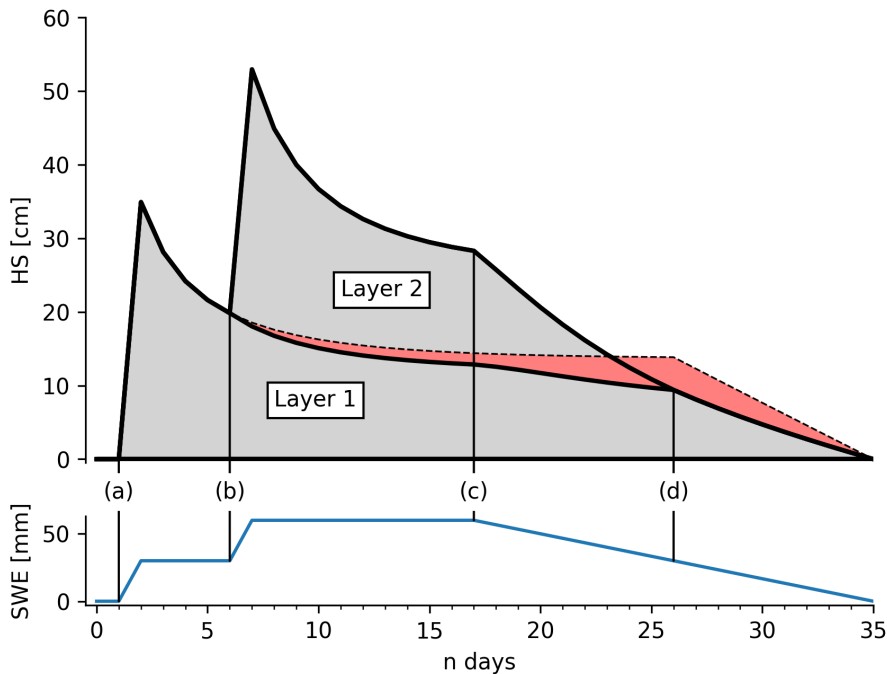

**Figure 1.** Illustrative snowpack evolution within the SWE2HS model calculated from 35 days of synthetic SWE forcing data. The top panel shows the calculated snow depth (top black solid line) and the two generated layers (grey areas) separated by a layer boundary (middle black solid line). The dashed black line represents the exponential settling for layer 1 according to Eq. 1 without the changes in $\rho_{max}$ due to the overburden from layer 2 and the decrease in SWE. The red area therefore represents the effects of changing overburden and wetting on the densification dynamics of layer 1. The letters on the x-axis indicate (a): Initial increase in SWE, layer 1 is created with density $\rho_{new}$ at creation time. (a)-(b): The density of layer 1 increases according to Eq. 1, $\rho_{max}$ is set to a higher value than $\rho_{max,init}$ since half the weight of layer 1 is treated as its overburden. (b): SWE increases and layer 2 is created. For layer 1, $\rho_{max}$ increases according to Eq. 2 and Eq. 3. (b)-(c): Both layers settle after Eq. 1 with the updated $\rho_{max}$ for layer 1. (c) SWE starts to decrease and $\rho_{max}$ starts to increase after Eq. 4 for both layers. (c)-(d): layer 2 is removed gradually in order to compensate for the decreasing SWE, both layers settle according to Eq. 1 and $\rho_{max}$ is updated for each time step according to Eq. 4. (d) layer 2 is completely removed. Subsequently, layer 1 is gradually removed in order to account for the further decrease in SWE.

initial value at creation time of the layer and subsequently increases towards a higher value based on the overburden a layer has experienced and the occurrence of SWE losses in the snowpack. When SWE decreases, the model removes layers from the top

95   of the snowpack. The layer number $n$ can thus undergo changes over time based on the number of SWE increases and losses in the snowpack. The model neglects constructive metamorphism, refreezing, and is not able to capture rain-on-snow events which might lead to an increase in SWE but no increase in snow depth.

## 2.2 Settling mechanisms

The density of a layer at day $i$ asymptotically converges towards the time-varying $\rho_{max}$ of the layer via the following exponential function:

$$\rho_i = \rho_{max} - (\rho_{max} - \rho_{i-1}) \cdot \exp(\frac{-1}{R}) \tag{1}$$

Where $\rho_i$ is the density of day $i$ and $\rho_{i-1}$ is the layer density of the day before. The settling resistance (e-folding time) $R$ is a model parameter which is optimized in model calibration.

The maximum density to which the density of a snow layer converges, $\rho_{max}$ in Eq. 1, also evolves over time. We model the maximum density of a snow layer based on three assumptions. The first assumption is that snow which has experienced a higher load reaches a higher maximum density. The second assumption is that a snow layer is initially dry and that wet snow has a higher maximum density than dry snow. The third assumption is that the time-varying maximum density cannot decrease. Accordingly, the maximum density of a snow layer undergoes changes during its lifetime and transitions from the model parameters $\rho_{max,init}$ to the model parameter $\rho_{max,end}$. At the time of deposition, the layer has a theoretical maximum snow density of $\rho_{max,init}$. Afterwards, $\rho_{max}$ increases towards $\rho_{max,end}$ by two mechanisms as described following.

1. If a layer experiences an overburden $\sigma > 0\,\mathrm{mm}$, its maximum density $\rho_{max}$ is increased linearly with overburden. We calculate $\sigma$ as a proxy for overburden stress by summing the amount of SWE above a layer and half of the SWE of the layer itself. If the overburden weight is equal or larger than the model parameter $\sigma_{max}$, $\rho_{max}$ is capped at $\rho_{max,end}$.

$$\rho_{max}^* = \begin{cases} \frac{(\rho_{max,end} - \rho_{max,init})}{\sigma_{max}} \cdot \sigma + \rho_{max,init} & \text{if } \sigma < \sigma_{max} \\ \rho_{max,end} & \text{if } \sigma \geq \sigma_{max} \end{cases} \tag{2}$$

If the updated $\rho_{max}^*$ is lower or equal than the value of the day before ($\rho_{max\ i-1}$), the value of the current day ($\rho_{max\ i}$) is not updated which could otherwise cause a decrease of $\rho$ if the density $\rho_{i-1}$ equals $\rho_{max\ i-1}$ (see Eq. 1).

$$\rho_{max,i} = \begin{cases} \rho_{max}^* & \text{if } \rho_{max}^* \in (\rho_{max\ i-1}, \rho_{max,end}] \\ \rho_{max\ i-1} & \text{if } \rho_{max}^* \leq \rho_{max\ i-1} \end{cases} \tag{3}$$

2. SWE losses are defined by $\mathrm{SWE}_i - \mathrm{SWE}_{i-1} < 0$. Whenever SWE in the snowpack decreases, we assume that the entire snowpack is wet since we attribute all SWE losses to runoff. In doing so, we neglect losses in SWE due to sublimation. If SWE decreases, we assume melt metamorphism is active and the maximum snow density $\rho_{max}$ of each layer is increased towards $\rho_{max,end}$ by

$$\rho_{max\ i} = \rho_{max,end} - (\rho_{max,end} - \rho_{max\ i-1}) \cdot \exp(-v_{melt}) \tag{4}$$

Where $\rho_{max\ i}$ is the maximum density of day $i$, $\rho_{max\ i-1}$ is the maximum density of the day before and $v_{melt}$ is a model parameter for the speed of that transition.

At the end of every time step, the snow depth of the snowpack is calculated by summing up the thickness of all $n$ snow layers in the snowpack as

$$\text{HS} = \sum_{k=1}^{n} \frac{\text{SWE}_k \cdot \rho_{water}}{\rho_k} \tag{5}$$

where $\rho_{water}$ is the density of water with $1000\,\text{kg}\,\text{m}^{-3}$ and $\text{SWE}_k$ and $\rho_k$ are SWE and density of layer $k$, respectively. All free model parameters that need to be calibrated are listed in Table 2.

## 2.3 Technical implementation

We provide an implementation of the model as a Python package under GNU General Public License v3.0 (GPLv3). One-dimensional station data and two dimensional model grids of daily SWE time series can be transformed to snow depth with the above described snowpack evolution. Additionally, a step by step processing mode with caching of the model state variables for two dimensional SWE grids of consecutive days is available for operational applications. Python, being a high-level, interpreted general-purpose programming language has been chosen due to its easy-to-read syntax, growing user base and community support for scientific computing and data analysis. Our implementation is using the just-in-time Python compiler *Numba* (Lam et al., 2015) for increasing runtime efficiency. Additionally, it depends on the libraries *NumPy* (Harris et al., 2020) for numerical computations, *Pandas* (Reback et al., 2022) for one dimensional input series, and *xarray* (Hoyer and Hamman, 2017) for multidimensional input grids. The multidimensional, distributed versions of the model can make use of *Dask* (Dask Development Team, 2016) which makes it possible to execute the model in parallel on standalone computers or high performance computing environments. Processing 23 years of daily SWE data from the Swiss 1 x 1 km domain (8401 x 365 x 272 pixels) generated with the method of Michel et al. (2023) took ~10 min on a desktop PC (8 cores, Intel Core i7-4790 CPU @ 3.60 GHz, 24 GB RAM) including file IO. The model implementation can be installed from the official third-party software repository for Python, The Python Package Index (PyPI: https://pypi.org/project/swe2hs, last access: 20.10.2022), the source code of SWE2HS is hosted on a Gitlab instance of the Swiss Federal Institute for Forest, Snow and Landscape Research WSL (https://code.wsl.ch/aschauer/swe2hs, last access: 20.10.2022), and the software version which was used for this publication is archived at https://doi.org/10.5281/zenodo.7228066 (Aschauer, 2022).

## 3 Model calibration and validation

As for every empirical model, parameters in our density model need to be calibrated. Calibrated parameters may differ depending on the station, snow type and snow climatological setting. Here, we try to find one single generic optimal parameter set which suits most snow climatological conditions in Switzerland and the European Alps in general. We do so by calibration over a data set which covers a large range of different altitudes and climatologic settings in Switzerland (see Section 3.3.1) and test the found parameters on two other independent data sets compiled from automatic weather stations in the European Alps (see Section 3.3.2) and from withheld data of the calibration data set which was not used for calibration.

## 3.1 Calibration and validation methods

Our model has 6 model parameters which need to be calibrated. Before calibration, we define upper and lower bounds of possible values for each model parameter (see Table 2) and apply the constraints that $\rho_{max,init}$ needs to be smaller than $\rho_{max,end}$ and $\rho_{new}$ needs to be smaller than $\rho_{max,init}$. The chosen upper and lower parameter bounds are based on literature (e.g. for new snow density $\rho_{new}$) or based on our previously gathered experience with the model for the model specific parameters such as the settling resistance $R$.

For parameter calibration, we use the Differential Evolution algorithm which is a stochastic population based method for minimizing nonlinear and non-differentiable continuous space functions as implemented in SciPy (Storn and Price, 1997; Virtanen et al., 2020). We chose Differential Evolution due to its gradient free nature and ability to overcome local minima (Storn and Price, 1997). After an initial Sobol' sequence sampling (Sobol', 1967), the algorithm draws parameter candidate samples from the parameter space by mutating the current best member of the sample population with the difference of two other randomly chosen members. After the global optimization with the Differential Evolution algorithm, the result is refined by the L-BFGS-B method of Byrd et al. (1995) which is a Quasi-Newtonian method that estimates the Hessian of the objective function based on the recent parameter sample history and can handle bound constraints.

We optimize the model by minimizing the root mean squared error (RMSE) which is a measure of the distance between the predicted values from the model $\hat{y}$ to the reference $y$. It is defined as

$$\text{RMSE}(y, \hat{y}) = \sqrt{\frac{1}{n} \sum_{i=1}^{n} (\hat{y}_i - y_i)^2} \tag{6}$$

with $y_i$ and $\hat{y}_i$ being the $i$-th element of the $i = 1, ..., n$ elements in $y$ and $\hat{y}$, respectively. Additionally, we use the two statistical error measures coefficient of determination ($R^2$) and BIAS in order to evaluate the model. The $R^2$ score is representing the proportion of variation in the data $y$ that can be predicted from the model and is defined as

$$R^2(y, \hat{y}) = 1 - \frac{\sum_{i=1}^{n} (y_i - \hat{y}_i)^2}{\sum_{i=1}^{n} (y_i - \bar{y})^2} \tag{7}$$

with $\hat{y}_i$ being the predicted value of the $i$-th sample, $y_i$ being the associated reference value for total $n$ samples and $\bar{y}$ being the mean of $y$. The BIAS is a measure for the systematic tendency of a model to over- or under represent the reference data. Therefore, it has large implications in climatological contexts. We calculate the BIAS for a sample of size $n$ as follows:

$$\text{BIAS}(y, \hat{y}) = \frac{1}{n} \sum_{i=1}^{n} y_i - \hat{y}_i \tag{8}$$

where $\hat{y}_i$ is the predicted value of the $i$-th sample and $y_i$ is the associated reference value. All presented score values RMSE, $R^2$ and BIAS are calculated only for the subset of $\hat{y}$ and $y$ where any of the two vectors is not zero.

## 3.2 Sensitivity analysis methods

In order to assess the importance of individual model parameters on the result, we perform a sensitivity analysis on the validation data set by calculating $R^2$ and BIAS on 114688 parameter sets sampled after the method of Saltelli (2002). This method

expands on the low-discrepancy quasi-random Sobol' sequence and generates uniformly distributed samples of the parameter hypercube space. Afterwards, we run the model with the sampled parameter sets on the automatic stations validation data set (see Sect. 3.3.2), calculate the error measures for $R^2$ and BIAS for each parameter set and compute global sensitivity indices ($S_{Ti}$) after Sobol' (2001). These indices give an estimate about the proportion of variance in $R^2$ and BIAS that can be attributed to a model parameter and all its interactions with other model parameters. We perform the sensitivity analysis within the Python framework *SALib* of Herman and Usher (2017).

## 3.3 Calibration and validation data

### 3.3.1 Data from Swiss manual observer stations

To calibrate the SWE2HS model, we use data from 58 Swiss manual observer stations between 1080 and 2620 m a.s.l. operated by the WSL Institute for Snow and Avalanche Research SLF (Marty et al., 2017). Snow depth is measured daily with a snow stake and SWE is measured every two weeks in a snow pit on the same site. In order to get daily SWE data, HS data is transformed to SWE with the $\Delta$SNOW model of Winkler et al. (2021). $\Delta$SNOW is a semi-empirical multilayer snow compaction model that uses a continuous time series of snow depth to calculate SWE and bulk snow density. The model calculates snow compaction by treating snow as a Newtonian viscous fluid, is able to include transient settlement due to new snow loading, and accounts for measurement errors in the input data. Once a defined maximum density is reached in a layer, the snow is melted and distributed top to bottom in the snowpack. Runoff is triggered when all layers reach the maximum snow density. The model has 7 free parameters which were calibrated in Winkler et al. (2021) with data from 14 stations at different elevations and snow climatological regions in the European Alps. The $\Delta$SNOW model performs very well in modeling the temporal evolution of SWE on the daily scale (Fontrodona-Bach et al., 2023). Additionally, the accuracy of the $\Delta$SNOW model is already high for the original unified Alpine-wide parameters as shown by Winkler et al. (2021) and confirmed by Fontrodona-Bach et al. (2023). In order to improve this accuracy of the daily SWE time series, the $\Delta$SNOW model parameters were optimized for each station individually using the biweekly SWE measurements from the manual observer profiles. Due to its destructive nature, the snow pit is not at the exact same location as the snow stake and consequently the profile cut height can deviate from the measured height at the snow stake. Therefore, the biweekly SWE data were corrected by calculating the bulk density from the profile and applying it to the measured height from the snow stake. $\Delta$SNOW model parameter optimization was done by minimizing the RMSE between modeled SWE and corrected SWE from the profiles while we allowed the $\Delta$SNOW parameters $\rho_{max}$ (maximum density) to vary between 300 and 600 kg m$^{-3}$, $\rho_0$ (new snow density) to vary between 65 and 135 kg m$^{-3}$, and the remaining parameters to vary by $\pm 25\%$ from the optimized value found in Winkler et al. (2021). For optimization, we again used Differential Evolution as described in Sect 3. In order to further increase the reliability of the calibration data set, we only kept station-winters with more than 2 SWE profiles and RMSE below 7.5 mm in the resulting daily SWE data set from the $\Delta$SNOW model. Since we did not want certain stations with long SWE and HS records to bias the calibration, we shortened the length of station records longer than 15 years by randomly selecting 15 water years from the full station record. The resulting set consists of 741 station-years from 60 stations. Compared with the biweekly manual SWE measurements, the modeled daily

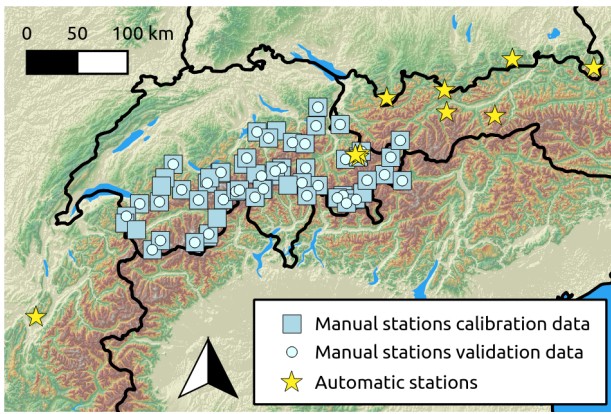

**Figure 2.** Locations of the snow measurement sites in the European Alps from which snow water equivalent and snow depth data was used to compile the calibration and two validation data sets. Background coloring resembles elevation. GMTED2010 elevation data, courtesy of the U.S. Geological Survey, was used to create the map (Danielson and Gesch, 2011).

SWE calibration set has an RMSE of 30.0 mm and BIAS of -1.09 mm in the calibration data set. We however cannot assess the uncertainty for the dates between the biweekly SWE measurements.

While we are aware that it might be preferable to calibrate a model on measured data instead of output from another model, we still chose the above described approach in order to have an exhaustive calibration data set which a) covers a wider range of altitudes, expositions and snow climatic settings in our target region, b) does not have problems of potential over- and under measurement from automatic SWE measurement devices (Johnson and Marks, 2004), and c) does not contain measurement noise which is not properly tackled in the SWE2HS model.

As a first validation data set, we use the remaining years of the long station records which were shortened to 15 water years for the calibration data set (see above). This calibration set of manual observer station data contains 1279 station-years from 42 different stations.

### 3.3.2   Data from automatic weather stations in the European Alps

As a second validation data set, we gathered data of 10 different automatic weather stations (AWSs) in Austria, France, Ger-
many, and Switzerland that automatically measure SWE with either a snow pillow or a snow scale and measure snow depth with an ultrasonic measurement device at sub-daily resolution (see Table 1). The raw SWE and HS data with a temporal resolution ranging between 15 min and 1 h were resampled to daily resolution by taking the median of all measurements between 6 a.m. and 8 a.m. local time. Any systematic offset errors from raw sensors were corrected by subtracting the mode of the summer months (MJJASON) from the SWE or HS time series of each hydrological year. Missing data gaps shorter than 5
consecutive days in SWE have been filled by linear interpolation. In total, 77 short SWE gaps were filled, including 39 1-day gaps, 17 2-day gaps, 8 3-day and 4-day gaps respectively, and 5 5-day gaps. For longer gaps, the time period before the gap in the respective hydrological year is included and data after the gap is discarded. Short gaps in snow depth are accepted since

**Table 1.** Automatic weather stations from which we used snow water equivalent and snow depth data for validation of the model. The number of years refers to complete hydrological years (Sep-Aug) included after data cleaning, the average snow depth ($\overline{\text{HS}}$) is calculated in the winter months from November to April.

| Site name | Source / data provider | Altitude [m a.s.l.] | #years of data | $\overline{\text{HS}}$ (Nov-Apr) [m] |
|---|---|---|---|---|
| Col de Porte (FR) | Lejeune et al. (2019) | 1325 | 13 | 0.51 |
| Davos (CH) | SLF[1] | 1563 | 1 | 0.48 |
| Fellhorn (DE) | LWZ Bavaria[2] | 1610 | 14 | 0.88 |
| Kühroint (DE) | LWZ Bavaria | 1420 | 13 | 0.75 |
| Kühtai (AT) | Krajci et al. (2017) | 1920 | 21 | 0.80 |
| Laret (CH) | SLF | 1513 | 2 | 0.69 |
| Spitzingsee (DE) | LWZ Bavaria | 1100 | 9 | 0.47 |
| Wattener Lizum (AT) | Hagen et al. (2023) | 1994 | 12 | 0.67 |
| Weissfluhjoch (CH) | SLF | 2536 | 12 | 1.35 |
| Zugspitze (DE) | LWZ Bavaria | 2420 | 9 | 1.64 |

[1]WSL Institue for Snow and Avalanche research SLF    [2]Bavarian avalanche warning center

**Table 2.** Parameters of the model, lower and upper bounds during calibration and optimized value.

| Parameter | Description | Unit | Lower bound | Upper bound | Optimized value |
|---|---|---|---|---|---|
| $\rho_{max,init}$ | Initial maximum density | $\text{kg m}^{-3}$ | 150.00 | 300.00 | 204.135 |
| $\rho_{max,end}$ | Final maximum density | $\text{kg m}^{-3}$ | 300.00 | 600.00 | 427.181 |
| $\rho_{new}$ | New snow density | $\text{kg m}^{-3}$ | 50.00 | 150.00 | 85.914 |
| $R$ | Settling resistance | - | 1.00 | 110.00 | 5.923 |
| $\sigma_{max}$ | Overburden where $\rho_{max,end}$ is reached | mm w.e. | 100 | 2000 | 227 |
| $v_{melt}$ | Melt metamorphism transition rate | - | 0.05 | 2.00 | 0.134 |

it is not required to drive the density model but for evaluating the quality of the model. Missing HS data points will thus not be included when calculating any score metrics. All hydrological years that were included in the final validation data set have been quality checked by visual inspection and by ensuring the bulk density stays below $700\,\text{kg m}^{-3}$.

## 4 Results

The model calibration on the data set described in Sect. 3.3.1 yielded the optimized parameters listed in Table 2. Fig. 3 shows the temporal evolution during 6 example winters for stations in the AWSs validation data set calculated with the optimized parameter set. Looking at the temporal development of the density and layering in the modeled snowpack, the density rapidly increases in the first few days after layer creation leading to enhanced settlement in this period. Additionally, the density of a

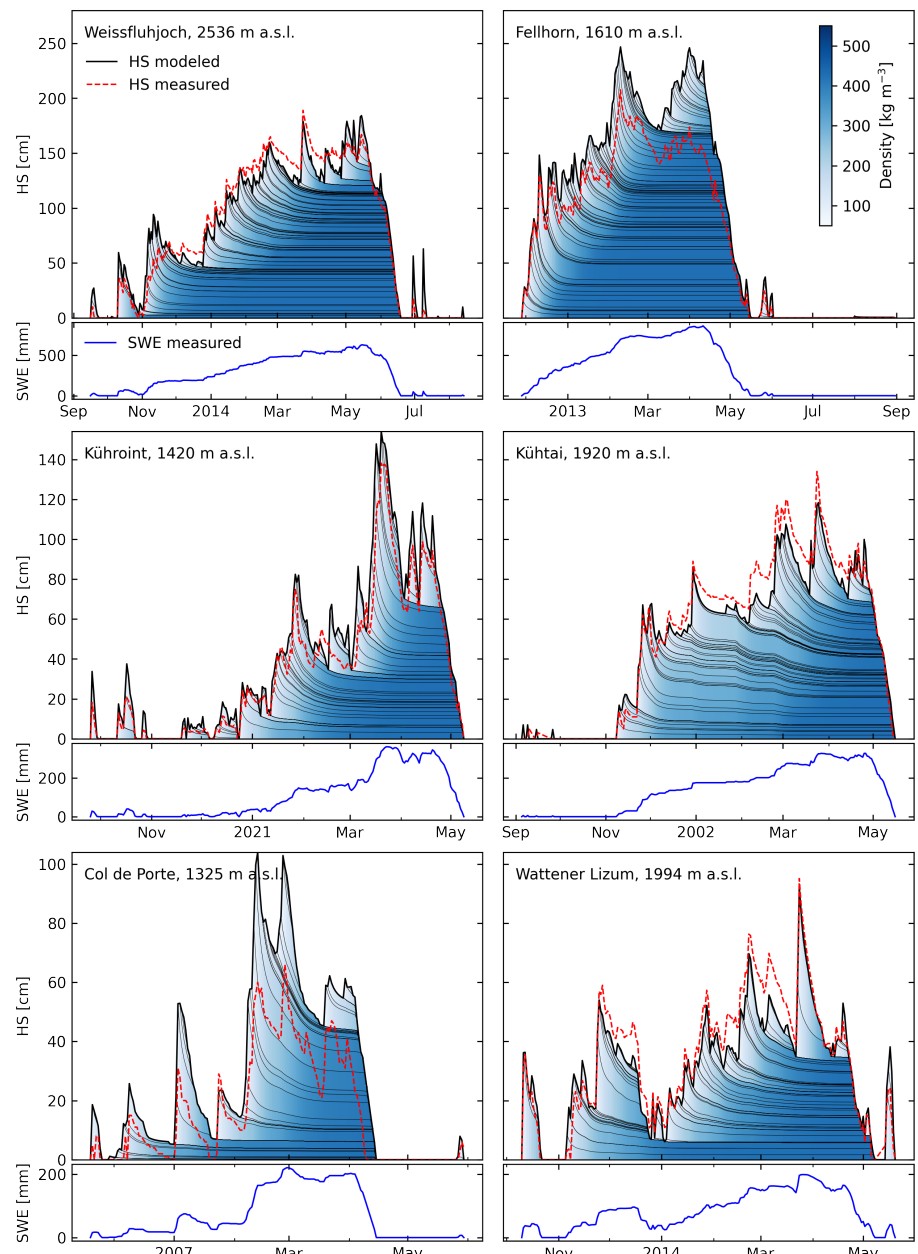

**Figure 3.** Schematic modeled snowpack evolution for 6 different station-years from the automatic stations validation data set of stations with different altitudes and snowpack thicknesses (see Table 1). In the top panels, the red dotted line is the measured snow depth (HS), the black solid line bounding the colored area is the modeled snow depth, the thin black lines represent the layer boundaries within the modeled snowpack, and the coloring refers to the modeled layer densities. The lower panels show the daily snow water equivalent time series used to force the model in each station year.

**Table 3.** Scores values RMSE, $R^2$ and BIAS for the calibration data set and the two validation data sets after parameter calibration. The accompanying data is visualized in Figs. 4 and 5.

|  | RMSE [cm] | $R^2$ | BIAS [cm] |
|---|---|---|---|
| Manual stations calibration data | 8.4 | 0.971 | -0.3 |
| Manual stations validation data | 7.9 | 0.971 | -0.3 |
| Automatic stations | 20.6 | 0.915 | 1.8 |

layer reacts to changes in SWE in the overlying layers (see e.g. bottom three layers, end of January 2021 for station Kühroint). For the deep snowpacks in the top two panels in Fig. 3, the lower layers are no longer able to respond to overburden and no further compaction occurs once a certain amount of snow is on top (see Eq. 2). The same settling dynamics in the snowpack can be observed for the calibration data set and manual stations validation data set (see Fig. A2 and Fig. A1).

With the optimized parameter set, the model is able to fit the calibration data with RMSE of 8.4 cm, $R^2$ of 0.97 and negligible BIAS of -0.3 cm (see Table 3 and Fig. 4, left panel). The seasonal evolution of the bulk density can be reproduced well on the calibration data set with monthly $R^2$ scores larger than 0.93 and June being the only month with considerable underestimation of the median snow depth (Fig. 5, left panels). On the manual stations validation data set, the model is able to achieve the same performance as on the calibration data set with RMSE of 7.9 cm, $R^2$ of 0.97, and BIAS of -0.3 cm. Monthly $R^2$ values
for the manual station validation data set are above 0.94 for the winter months of November through May. The $R^2$ values are lower for the months of October and June with 0.88 and 0.89, respectively. On the AWSs validation data set, the performance is weaker than for the calibration data set and manual station validation dataset with RMSE of 20.5 cm, $R^2$ of 0.92 and BIAS of 2.5 cm. The model slightly underestimates the median snow depth in February and March on the validation data set and overestimates the median snow depth in the ablation months April, May and June. Monthly $R^2$ score values are above 0.79 for
all months except of October where $R^2$ is below 0.4. On the calibration data set, $R^2$ is for 75% of the stations above 0.95, only for two stations $R^2$ is below 0.8 (see Fig. 6). The per-station $R^2$ scores in the manual stations validation data set are similarly distributed as for the calibration data set but for no station, $R^2$ score is below 0.8. On the validation data set, $R^2$ per station varies between 0.15 and 0.95 and is larger than 0.75 for 75% of the stations. On the calibration data set, the BIAS per station is uniformly distributed around 0 and for all except of two stations smaller than $\pm 10$ cm. On the validation data set, the BIAS
per station is ranging from -7 cm to 22.7 cm with four stations having a positive BIAS larger tan 10 cm.

According to the sensitivity analysis, the settling resistance factor $R$ is the most important model parameter with a global sensitivity index of 0.44 and 0.43 for $R^2$ and BIAS respectively (Fig. 7). This means, that within the 114688 samples drawn during the sensitivity analysis, 44% and 43% of the proportion of variance in in $R^2$ and BIAS can be attributed to the settling resistance factor $R$, respectively (see Sect. 3). For $R^2$, the second most influential model parameter is new snow density $\rho_{new}$
followed by the final maximum snow density $\rho_{max,end}$. For BIAS, the model is less sensitive to $\rho_{new}$ than for $R^2$. The model is relatively insensitive to changes in the model parameters $rho_{max,init}$, $v_{melt}$, and $\sigma_{max}$ with total sensitivity indices below 0.1 for both $R^2$ and RMSE.

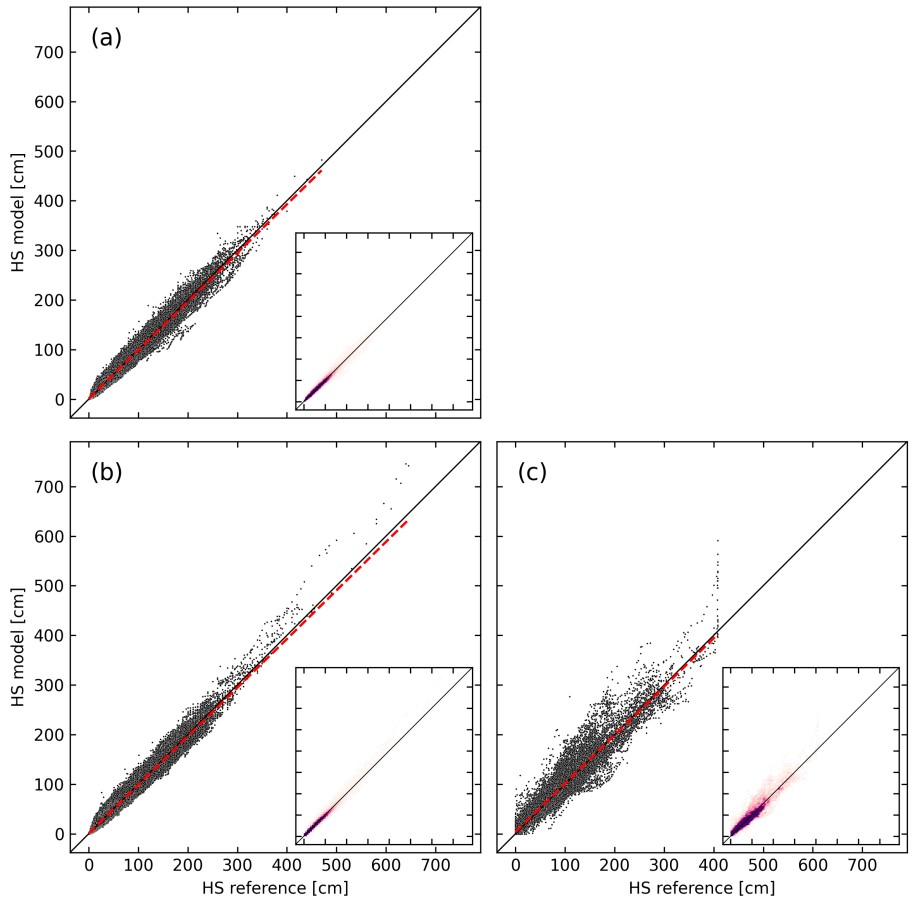

**Figure 4.** Scatterplots of modeled against measured snow depth values for (a) the calibration data set from Swiss manual observer stations, (b) the validation data set of the remaining years from the Swiss manual observer stations which were not used for calibration, and (c) the validation data set from automatic snow stations. The red dashed line is a linear fit to the data, the black solid line represents perfect predictions. In the insets, the same data is shown as bivariate histograms indicating the density of the scatter points. Score values of the shown data are listed in Table 3.

# 5   Discussion

## 5.1   Model design and complexity selection

On our way towards the model presented here, we tried models of different complexity, included and removed processes while iterating back and forward. Some prototype model versions additionally included daily temperature as input forcing, which we tried to use for parametrization of new snow density and onset of the wetting from top of the snowpack by using the cold content parameterizations used in Scheppler (2000) and Szentimrey et al. (2012). Other versions parametrized the settling resistance $R$ based on overburden or density of a layer or a combination of the two factors. In order to do an objective model

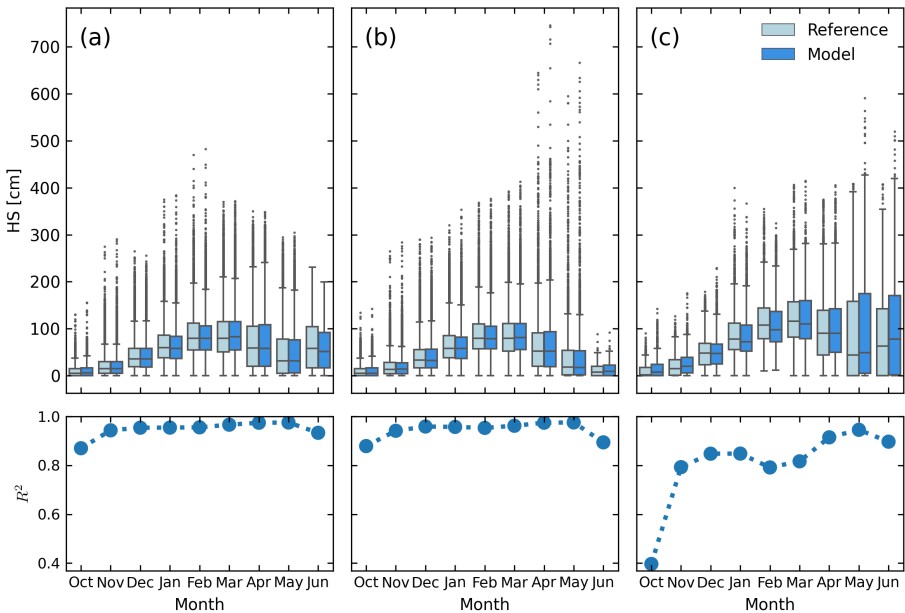

**Figure 5.** Boxplots comparing the distributions of measured and modeled data in the months from October to June for (a) the calibration data set from Swiss manual observer stations, (b) the validation data set of the remaining years from the Swiss manual observer stations which were not used for calibration, and (c) the validation data set from automatic snow stations. A box spans the lower and upper quartile of the data with a line at the median. The whiskers extend to last datum within 1.5 times the interquartile range while the points represent outliers past the range of the whiskers. The lower three panels show monthly $R^2$ scores for the modeled data to the reference, calculated for each data set.

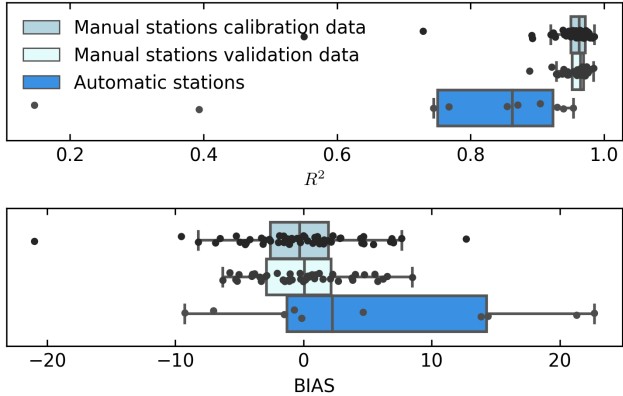

**Figure 6.** Boxplots of the scores $R^2$, and BIAS calculated individually on the data from each station in the calibration and two validation data sets. The black dots show the underlying data from which the boxplots were calculated.

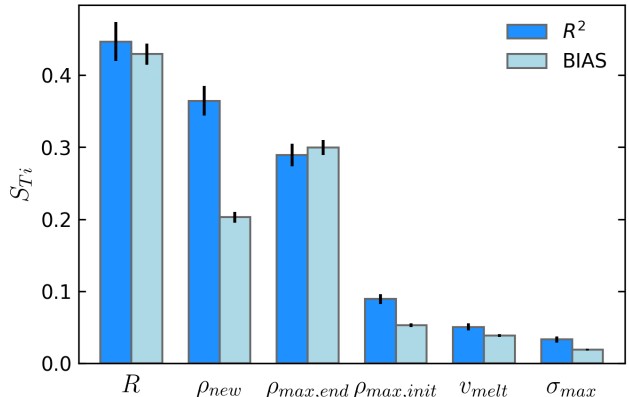

**Figure 7.** Global Sobol' (2001) sensitivity indices ($S_{Ti}$) calculated for $R^2$ and BIAS of the model predictions on the automatic stations validation data set from 114688 samples drawn with the method of Saltelli (2002).

selection for a final model version, additionally to the scores defined in Sect. 3 we calculated the Akaike Information Criterion (AIC) by using the RMSE as an estimator for the maximum value of the likelihood function. The AIC is a statistical error measure that penalizes larger numbers of free model parameters (Akaike, 1998; Cavanaugh and Neath, 2019). We then ranked the 4 different scores for the optimized model of each version and averaged the score ranks over the calibration and validation data set. This allowed us to make an informed decision on which model version to use. In order to avoid overfitting on the calibration data set, we gave more focus on the validation data set.

We assumed it would be beneficial to use temperature in the beginning of model conceptualization and one could argue, that when using SWE from an accumulation and ablation model, there is always at least daily mean temperature available used to drive a melt model. However, when quantitatively assessing the model versions with parametrized new snow density or cold content parameterizations we did not see model improvement from the daily mean temperature inclusion and thus decided to only use SWE. This additionally comes with the asset that the model can be plugged in as a post-processing tool to any snow model which outputs daily SWE. Besides the best performance, another important factor to keep the model simple was to reduce the risk of equifinality, meaning that an optimal solution can be achieved through different states i.e. parameter combinations of the model (Beven and Freer, 2001). In a very early version of the model, we had implemented 3 modes of layer removal: from top, proportional, and from bottom. In a deep snowpack, we assume most melting occurs at the top due to energy input from short and long wave radiation. There is also secondary melting due to geothermal heat fluxes, but we assume that this is negligible compared to the energy input from top. During melting, SWE is not actually "removed" from the top of the snowpack, unless it is sublimating directly into the air. In fact, melted liquid water is assumed to percolate down through the snowpack where it can potentially refreeze if the snowpack has not yet become isothermal. However, in a simple empirical model such as SWE2HS, it is not possible to implement such processes in a meaningful way and for shallow snowpacks, other processes and predominant melt mechanisms might be relevant. Nevertheless, predominant melting at the top of the snowpack for deep snowpacks could imply that the top layer should be removed first. These considerations, together with the fact that

the layer boundaries looked most realistic in comparison to measured settlement curves for both shallow and thick snowpacks, led us to the choice of layer removal from the top presented here. This has the effect, that new snow events late in the season followed by an immediate decrease in SWE, have a short-lived effect on the bulk snow density. We would also assume, that the top-down removal of layers reduces the sensitivity of the model to new snow density in general.

## 5.2 General remarks and limitations

As shown in Sect. 4, the model is able to fit the calibration data very well. The calibration data has been compiled from manually measured snow depth data and modeled SWE data with the $\Delta$SNOW model of Winkler et al. (2021). Therefore, e.g. the occurrence of rain-on-snow events cannot degrade the model skill since the $\Delta$SNOW model is also not able to represent rain-on-snow events. We still consider using modeled data for calibration a valid approach as we hold back a set of measured data for independent validation of our optimized model parameter set (see Sect. 3.3.2). Additionally, SWE2HS' main scope of application is post-processing output from simple accumulation and melt models. The model is performing better on the $\Delta$SNOW data set compared to the AWSs data set. This is due to several reasons. First and foremost, the model has been calibrated on the data in the $\Delta$SNOW data set and not on the data in the AWSs data set. Accordingly, the fitted parameters do not necessarily suite the data in the AWSs data set. Another reason is that the calibration and AWSs validation data sets cover different spatial domains. The scores on the manual stations validation data set, which are comparable to those on the calibration data set, show that the model in principle is able to perform on unseen data not used for calibration. Another potential reason why the scores on the AWSs validation data set might differ from those on the calibration data set is the fact, that the calibration of the SWE2HS model is made with modeled data from the $\Delta$SNOW of Winkler et al. (2021) (see Sect. 3.3.1). This modeled input data is also attributed with a considerable uncertainty compared to the biweekly manual SWE data (RMSE of $30.0\,\mathrm{mm}$) which can influence the calibrated parameters in a way that degrades the model skill on the AWSs data set. Other reason for weaker model performance in the AWSs data set are potentially arising from noise and measurement uncertainties. One source of these uncertainties are problems of over- and under measurement from the automatic SWE measurement devices (Johnson and Marks, 2004). This uncertainty increases with time during the winter and could be an explanation for the overestimation in the ablation season. Additionally, the measurement uncertainty of the automatic SWE and HS data can cause small changes in SWE and HS, which are not physically based (Capelli et al., 2022). In this regard the SWE and HS in the calibration data set is much more consistent. We could not include a mechanism to deal with measurement uncertainties analogous to Winkler et al. (2021) since a SWE time series does not contain any information on settlement which could be used to correctly distinguish a signal from noise. A last source of uncertainty in the AWSs data set is that the automatic SWE measurements are not necessarily located at the exact same place as the snow depth measurements and we did not have a way to correct this error in the same way as we did for the manually observed SWE and HS measurements. The low $R^2$ score in October for the AWSs data set (see Figure 5) could be explained by the increased importance of new snow density in the beginning of the snow season. In this data set, October snow cover is often characterised by ephemeral snowfall events, where new snow density is more important due to the lack of settled older snow layers. New snow density can have a high variability (see e.g. Helfricht et al., 2018), which is not predictable if new snow density is kept fixed or independent of temperature or similar, as is done in the SWE2HS model.

Other sources of uncertainty are due to inherent limitations of our empirical modeling approach. As mentioned above, the model is not able to represent rain-on-snow events. In the exemplary snowpack evolution of the winter 2020/21 at station Kühroint, an increase in SWE causes modeled snow depth to increase although the measured snow depth is constantly decreasing during this time (mid of March 2021, Fig. 3, middle left panels). This could be an example of either erroneously

measured SWE or a rain-on-snow event which caused an increase in SWE but not in HS. The latter seems more likely since simultaneously to the increase in SWE, settling is enhanced for the measured HS and (as additional not-shown data demonstrates) the temperature is rising above $0\,°C$ in combination with precipitation. Morán-Tejeda et al. (2016) show that such events are rare and contribute to maximum $100\,mm$ for elevations above $2000\,m$ a.s.l. With a changing climate, rain-on-snow events might become more likely above $2000\,m$ a.s.l. but might decrease for low altitudes as a decrease in rainfall and shorter

snow cover duration are thought to counteract increased temperatures. In the SWE2HS model, we assume that the snowpack is completely wet when SWE decreases. However, prior to complete wetting, a wetting front propagates through the snowpack from top to bottom over time (Marsh and Woo, 1984). The choice to use only SWE as a forcing leads to the limitation that we can only detect the wetting front that reaches the bottom, which will be observable as a negative change in SWE. Therefore, the model likely misses the onset of melt metamorphism in the upper snowpack and overestimates HS during this time. In a

post-processing setting, this could be addressed by looking forward in time and assuming wetting prior to the SWE decrease, but this design choice would make it impossible to use the model in an operational setting when no information about future snowpack evolution is available. We try to partly compensate for this flaw by increasing the maximum density with increasing overburden. As $R^2$ is not decreasing in spring (Fig. 5) for the calibration and validation data set, the model seems to be able to predict snow depth during the ablation period reasonably well nevertheless. By assuming that the entire snowpack is wet when

SWE decreases, we neglect processes other than runoff that lead to decreases in SWE. Sublimation, the direct phase transition from solid water to water vapor, is one of these processes that can have a remarkable impact on the alpine water budget, especially at wind exposed high elevations and in forested areas Strasser et al. (2008). The inability to represent sublimation in the SWE2HS model can lead to an underestimation of snow depth, because SWE losses lead to an increase of the maximum snow density ($\rho_{max}$) in Eq. 1. This limitation should be kept in mind when applying the model to conditions where sublimation

processes play an important role.

Since the model is of empirical nature, the parameter set which is presented here for the European Alps might not be suited for other regions on earth with different climatologic conditions. If applied to other regions, the model parameters need to be calibrated again. However, as we never tested the model in e.g. Arctic regions, we cannot make any statements if the model is able to represent settling dynamics in these snow climatologic conditions even if it would be calibrated on data from there.

Although the calibrated parameter set presented in this paper is thought to be representative for the European Alps, it can be quite off for some stations in the validation data set with BIAS of up to $22.7\,cm$ (see Fig. 6). This high positive BIAS is occurring at station Davos (only one year of data) and station Laret (two years of data). We suspect that in all three years a short period of potential positive measurement errors from the SWE sensor caused the snow cover evolution to become defective. In order to achieve the best model skill for a single location, recalibrating the model to data from that location is necessary.

Simple empirical models that try to conceptualize processes in a non-physical way are often subject to the risk of potential model equifinality (Beven and Freer, 2001). While we chose to present a single calibrated parameter set in Sect. 4, there is still the risk that other parameter sets might lead to equally good predictions. However, an in-depth analysis of potential equifinality is out of scope of this model description paper and might be subject of future work.

     The model is less sensitive to changes in the model parameter $\rho_{new}$ (new snow density) for BIAS than for $R^2$. This is likely
due to the nature of the two statistical error metrics used. $R^2$ is measuring the proportion of variance the model is able to explain in the data and the BIAS is measuring if the model is on average under or overestimating the data to be predicted. The new snow density is mainly affecting the model result for the times shortly following increases of SWE and thus is not as important for model BIAS as for $R^2$. Due to the sensitivity to new snow density and the fixed new snow density approach in the model it is not reasonable to derive climatologic indices related to the amount of new snow such as the maximum increase
in HS during three days from model output.

     The SWE2HS model is tailored for use with daily resolution SWE data. When attempting to use the model with higher temporal resolutions such as hourly, additional processes to those considered in the model become increasingly important and additional parameters such as radiation and temperature are likely to be required to satisfactorily represent densification. For example, the new snow density will be much more variable on shorter time scales, and it is likely that the fixed new snow
density approach used in the SWE2HS model will not be sufficient at hourly resolution. In addition, the empirical transition rate from dry to wet snow ($v_{melt}$) would have to be changed when adapting the model to higher temporal resolutions.

## 6   Conclusions

We present a simple snow density model which can be used to transfer continuous daily snow water equivalent data to snow depth. The empirical multi-layer model uses exponential settling equations, a fixed new snow density and assumes a chang-
ing maximum snow density over time based on overburden and SWE losses. The model was calibrated with a gradient free evolutionary algorithm on a data set from the Swiss Alps that was generated from biweekly SWE and daily HS records. Prior to calibration, the biweekly SWE records were converted to daily values with the $\Delta$SNOW of Winkler et al. (2021). On the calibration data, the model is able to reproduce the measured snow depth with RMSE of 8.4 cm and BIAS of -0.3 cm. SWE2HS is validated on multi-year data from 10 automatic snow stations between 1100 and 2500 m a.s.l. in the European Alps where
it can reproduce the measured data with RMSE of 20.5 cm and BIAS of 2.5 cm. In addition, the model is validated against withheld data from the Swiss manual observer station data set not used for calibration, on which it achieves as good results as on the calibration data set, with RMSE of 7.9 cm, $R^2$ of 0.97, and BIAS of -0.3 cm. Due to its simplicity, SWE2HS can be used for climatological use cases where input data for more sophisticated densification models is sparse. Since the only input needed to drive the model is daily SWE, it can be also used to post-process model output from any other snow model or to
transfer SWE data obtained from automatic SWE sensors.

*Code and data availability.* The current version of the SWE2HS model source code, including documentation and examples is available at https://code.wsl.ch/aschauer/swe2hs (last access: 04.06.2023) and a Python package is available through PyPI at https://pypi.org/project/swe2hs/ (last access: 04.06.2023). The exact version that was used for this manuscript (v1.0.3) is archived at https://doi.org/10.5281/zenodo.7228066 (Aschauer, 2022). The data from Kühtai is described in Krajci et al. (2017) and is available from https://doi.org/10.5281/zenodo.556110. The data from Col de Porte is described in Lejeune et al. (2019) and is avaliable from https://doi.org/10.17178/CRYOBSCLIM.CDP.2018. The data from Wattener Lizum is avaliable from https://doi.org/10.5281/zenodo.7845618 (Hagen et al., 2023). The complete data sets used for calibration and validation, including coordinates and used parameters for the $\Delta$SNOW model for each station in the manual Swiss station network are available from https://doi.org/10.16904/envidat.394 (Aschauer and Marty, 2023). The code to generate all figures except of Fig. 2 is archived at https://doi.org/10.5281/zenodo.8002941 (Aschauer, 2023).

## Appendix A: Additional Figures

*Author contributions.* JA compiled the calibration and validation sets, developed the methodology and software code, and wrote the initial paper draft. CM, TJ and AM gave input to the methodology and reviewed different model versions. CM acquired funding and supervised the study. All authors reviewed and commented on the manuscript.

*Competing interests.* The authors declare that they have no competing interests.

*Acknowledgements.* We want to thank the Bavarian avalanche warning center (Lawinenwarnzentrale Bayern) in Munich and the Austrian Research Centre for Forests BFW in Innsbruck for contributing data from their measurement stations to the calibration data set. This work was financially supported by the internal project "Climatological maps for snow depth" of the Swiss Federal Institute for Forest Snow and Landscape Research WSL.

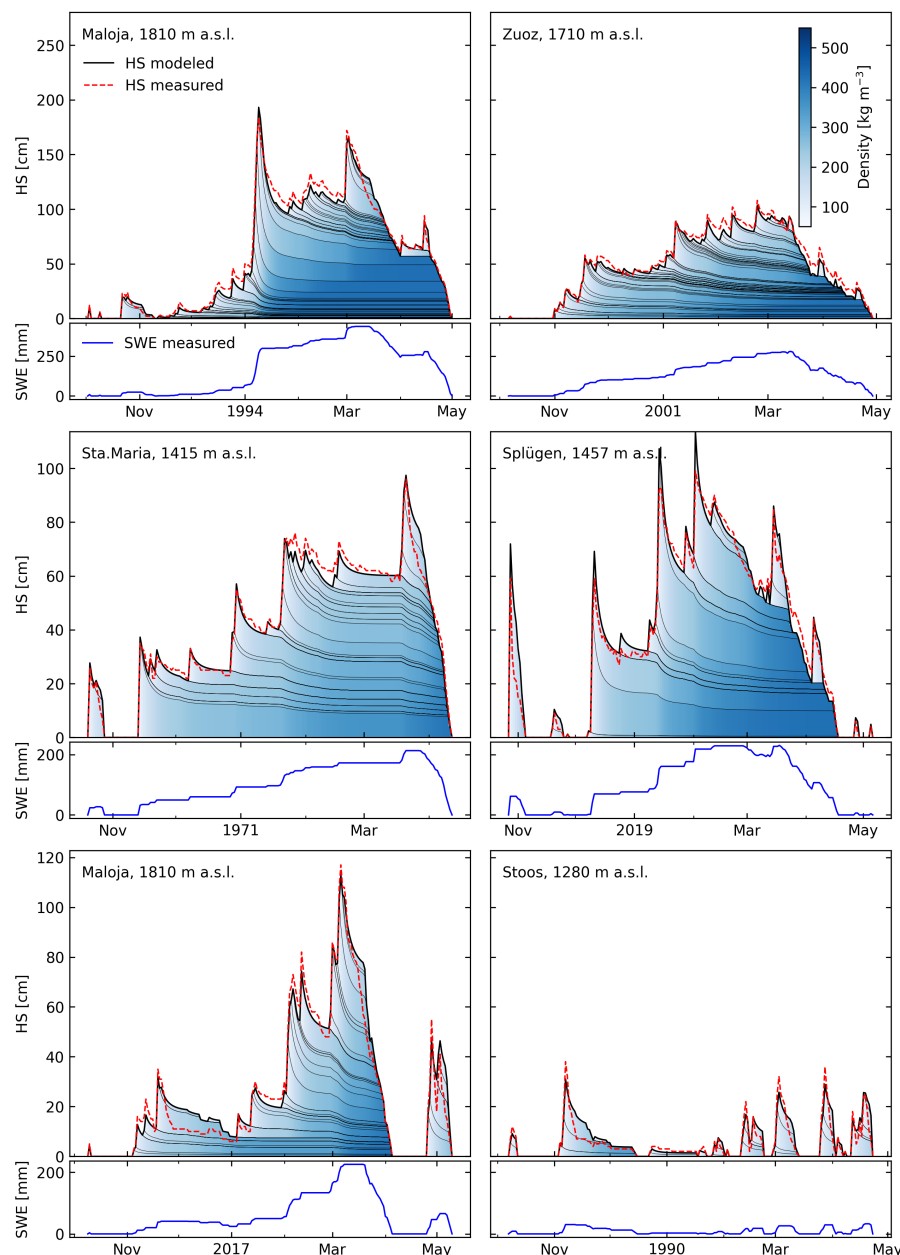

**Figure A1.** Schematic modeled snowpack evolution for 6 different station-years from the manual stations validation data set. Winters from stations with different elevations and with differing snowpack thicknesses are shown. For an explanation of the figure, the reader is referred to the caption of Figure 3.

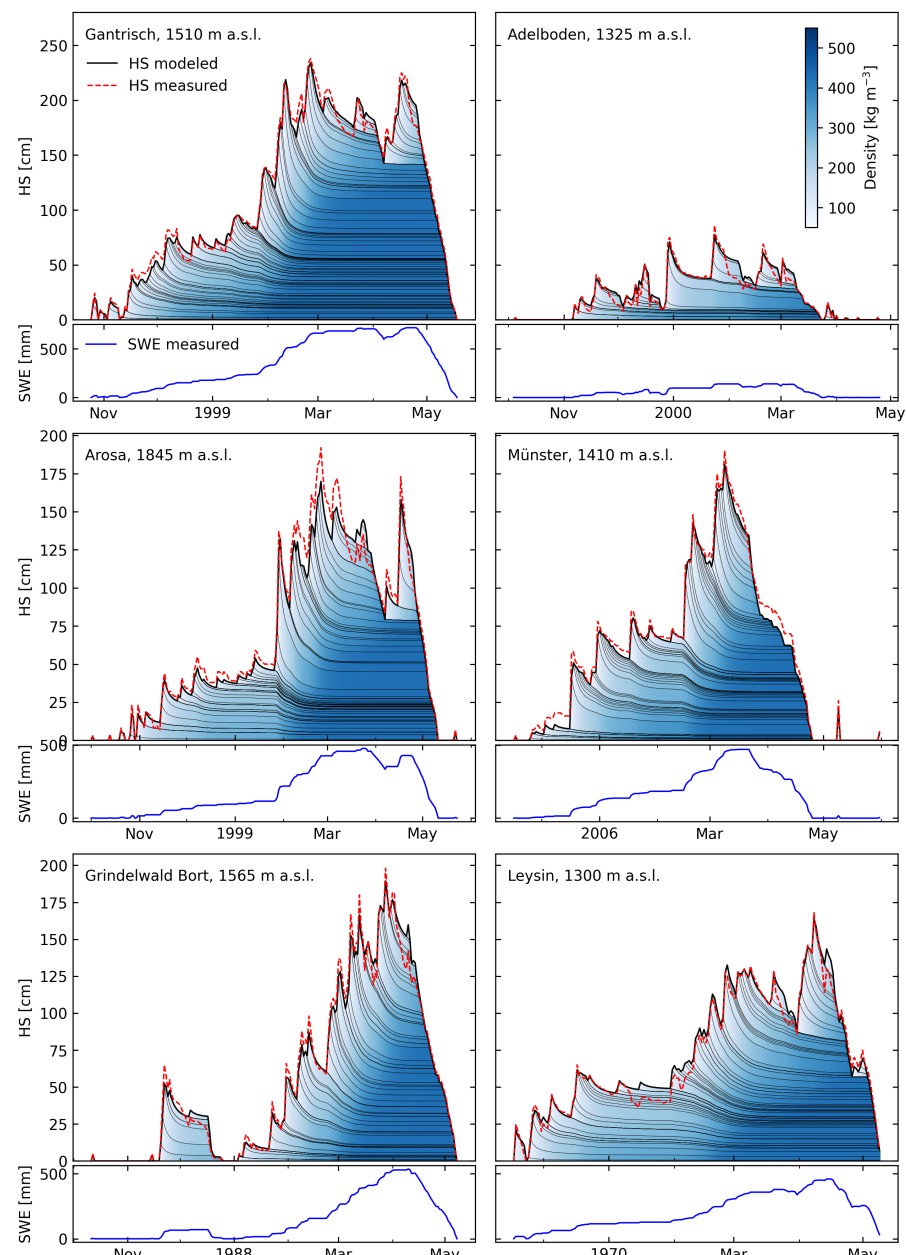

**Figure A2.** Schematic modeled snowpack evolution for 6 different station-years from the manual stations calibration data set. Winters from stations with different elevations and with differing snowpack thicknesses are shown. For an explanation of the figure, the reader is referred to the caption of Figure 3.

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
