# Peer review of "An empirical model to calculate snow depth from daily snow water equivalent: SWE2HS 1.0"

_Geoscientific Model Development, 2022_

## Author Comment (AC1)

**Author responses to comments RC1 and RC2 on gmd-2022-258 and revision summary**

We thank both reviewers for their time to assess our work and for the valuable feedback and suggestions. In this document, we summarize the changes for the revised manuscript and respond to each suggestion and comment from each reviewer one by one. The reviewer comments are highlighted in blue while our responses and comments are kept in black.

**Replies to comments from Anonymous Reviewer #1**

**Summary**

The manuscript 'An empirical model to calculate snow depth from daily snow water equivalent: SWE2HS 1.0' by Aschauer et al. provides a novel empirical model approach to convert daily snow water equivalent into snow depth in a simple way without having to rely on physically-based complexity or additional variables. The model uses a multi-layer densification approach based on exponential settling and takes changing maximum snow densities over time as a function of overburden and SWE losses into account. SWE2HS was calibrated on a dataset of the Swiss manual observer station network at 58 locations and was validated with 10 automatic weather stations in the European Alps. The chosen objective functions RMSE, R² and BIAS show good results for the calibration data set and quite good results for the validation data set, however, in the latter case, the RMSE is more than double as high (8.4 cm vs. 20.5 cm). I believe this model presentation is interesting to the readers of the journal as well as to the snow-hydro community. I see an advantage in using SWE2HS as a post-processing tool for HS conversions especially for SWE measurement devices and conceptual hydrological models, which simulate SWE instead of HS. In general, the manuscript is well written. However, it needs some clarifications before considering it for publication. The methods are valid, but need in some parts a better description. Please see my comments below.

**General comments**

- Please give more information on your model SWE2HS in the abstract, which is currently fully missing in this version of the abstract. You only mention that you developed SWE2HS and that it has been calibrated and validated; but important information on the characteristics of the model are missing (e.g., multi-layer densification; solely based on SWE; exponential settling; changing of maximum snow density over time due to overburden and SWE losses).

This was totally missing and we admit it is very important to get the basics of the model just by reading the abstract. We added a short description of the basic principles of the model to the abstract (Lines 7-9).

- In the Introduction section, you distinguish between empirical and semi-empirical models (l. 42ff). In the manuscript, your SWE2HS model is sometimes described as empirical (e.g. in the title) and sometimes as semi-empirical (esp. in the Discussion and Conclusions sections). Please be consistent and define whether your model is empirical or semi-empirical. According to your definition in the introduction, it is rather empirical.

Thank you for pointing that out. You are correct in classifying the SWE2HS model as empirical and, as the title suggests, so do we. Accordingly, we have changed the misleading parts in the discussion and conclusion to be consistent with the rest of the manuscript (Lines 336 and 389)

- In the current version, your model is for operation on a daily base. Would the model also work with hourly values and if not, what would you need to change in your model?

We think in order to adequately represent settling dynamics on a higher temporal resolution such as hourly, additional processes to the ones now considered become increasingly important. Additional variables like radiation and temperature are then probably needed to successfully transform SWE to HS in a satisfactory manner.

One big challenge is that the new snow density will be much more variable on shorter timescales, and one has probably to abandon the fixed new snow density approach used in the here presented SWE2HS model. Additionally, the empirical transition speed from dry to wet snow would need to change if one wants to adapt the model for higher temporal resolutions.

Further research could nevertheless try to tweak the here presented SWE2HS model for other timescales but we think this is out of scope for this publication.

We added a paragraph to the discussion which picks up the topic of temporal resolution and our thoughts on it (Lines 381-386).

- In Section 3.1, you apply the 'delta'-snow model by Winkler et al. (2021). Please introduce at least the main characteristics of this model.

We added a short description of the delta-snow model by Winkler et al. (2021) to Section 3.3.1 (Lines 196 - 204).

- The manuscript would largely benefit from a figure illustrating the main steps of the density model / settling mechanism described in Section 2.2 in a schematic (see for example Figure 1 in Winkler et al. (2021) although you have less processes involved than the 'delta'-snow model).

The figure below illustrates the basic concepts of compaction in the SWE2HS model using a two-layer synthetic snow pack evolution over 35 days. In this figure, we compare the settling in SWE2HS with a simpler model where the effects of overburden and wetting are neglected. We have added a detailed caption to the figure in the manuscript where we go step by step through the processes involved (new Figure 1 in the Manuscript)

[Figure]

• The separation into calibration and validation datasets seems to be a bit unequal. Why didn't you chose to validate for example a quarter of the dataset of the Swiss manual observer station network in addition to the automatic weather stations? It would be interesting to see how the validation performs on the Swiss manual observer station network as well. This would maybe overcome a bit the issue that you rely on the 'delta'-snow model solely in the calibration dataset and do not use it also for validation. I would suggest to have two validation data sets. The one you already have with the automatic weather stations and a second one with parts of the Swiss manual observer station network. For the latter you could also use additional years, which you cut off at some stations for the calibration.

Thank you for this suggestion. As you proposed, we now use the additional years from the Swiss manual observer station records longer than 15 years for a second validation data set. We adapted the Figures 4, 5, 6 and Table 3 in the results section in order to show results from the three different data sets and adapted the text in order to reflect the third data set.

• In addition to Figure 1, showing an example of the model performance for one year at the station Kühroint, it would be interesting to see also further examples, e.g. for other locations, higher laying stations with a deeper snowpack, your calibration data etc. (this could be shown also in a supplementary).

We have added five more station-years from the automatic stations calibration data to the figure, showing shallower and deeper snowpacks and cases where the model did not perform as well as for the Kühroint station in the winter of 2020/21.

In addition, we show 6 exemplary station-years for manual station calibration and manual station validation data, respectively (see Figures A1 and A1 in the Appendix).

**Specific comments**

- 19: Although it might be clear for most readers in the snow-hydrological community what you mean with temperature-index models, please shortly explain and insert a reference.

We added a brief description of temperature index modeling and a reference to Hock (2003) (Line 26).

- 37f: The sentence sounds a bit strange. I would reformulate to: 'Various parametrizations exist and are usually based on estimating new snow density as a function of wind speed, temperature and relative humidity (references).'

You are right, we changed the sentence according to your proposed reformulation (Line 47ff).

- Section 2 'Density model': For a better orientation for the reader, it would make sense to insert a (short) sub-chapter to introduce the general concept of the density model and not only present the 'Settling mechanism' in a sub-chapter. As mentioned in the general comments, it would be helpful to insert a schematic figure of SWE2HS.

We already introduce the general concept of the density model in the introductory paragraph of Section 2. In order to make this more clear, we inserted a new subsection heading in the beginning of Section 2 named "2.1 Basic concept". There, we also inserted the new schematic figure illustrating the main steps of the model (see answer above).

- 78: 'increases with an exponential decay function' instead of 'increases exponentially'

Changed accordingly (Line 92).

- 80: Why does the model just remove SWE from the top layer when SWE decreases (this is counter intuitive (compared to reality) and needs clarification)? Would it be an option that SWE is removed from each layer proportionally?

Thank you for this question. We added a few sentences to the discussion explaining our decision and discuss potential implications (Lines 293-305). In a very early version of the model, we had 3 modes of layer removal implemented: from top, proportional, and from bottom. Unfortunately, we did not base our choice of the "from top" method on score numbers from the different versions which we could use to defend our choice for this method. Our impression at the time was that the "from bottom" and "proportional" methods produced unrealistically looking layer boundaries when qualitatively compared to measured settling curves as in e.g. Steinkogler et al. (2009). The "from top" method produced the most realistic looking layer boundary evolution. Later on, we did not include the "proportional" and "from bottom" versions in the AIC model selection because we discarded these versions immediately.

In order to be more clear and to prevent misinterpretations, we rephrased the description of this part of the model in the manuscript and now remove "layers" instead of "SWE" from top of the snow pack (Line 94).

- 93: 'the' is written twice. Delete it one time.

Done.

- 102: 'which' is written twice. Delete it one time.

Done.

- 104ff: You state that you neglect sublimation and that the snowpack has become wet entirely when the snowpack is decreasing. Please add a short discussion on these two simplifications, which are / might be different in reality and potential sources of errors due to these two assumptions in your discussion section.

We have added another part to the discussion where we discuss the implications of the missing sublimation process in the model (Lines 354-360).

In the initially submitted discussion manuscript, we already wrote about the inability to track the wetting front which is propagating top to bottom prior to complete wetting (Lines 275-279). This is also related to the assumption that the snowpack has become wet entirely when SWE decreases. We rephrased the respective paragraph and specified the implications (potential overestimation of HS during wetting front propagation) more explicitly (Lines 345-352).

- 127: It is helpful to describe the computation time and it seems to be quite fast. Which dataset did you chose for this exercise; was it a SWE output of a snow-hydrological model? If you performed such a potential application of your model, it would be valuable to report on that in more depth (in a separate section).

In this benchmark exercise we used a daily 1km gridded SWE data set covering entire Switzerland and 23 years as described in Michel et al. (2023). In the SWE2HS model description paper, we want to focus on the methodology and calibration of the model. Describing potential applications is in our opinion out of scope of this manuscript.

Nevertheless, we put in a reference to Michel et al. (2023) and mention the potential application for a snow climatology map in Switzerland (Lines 75-78 and Line 142).

- Section 3 'Model calibration and validation': It would make sense to insert for better orientation for the reader a sub-chapter for lines 135-166, e.g., 'Calibration and validation methods' before introducing the data sets in separate sub-chapters.

We inserted a subsection named "Calibration and validation methods" which includes the first part of Section 3.

- 139f: It is not yet clear how you set the upper and lower bounds of possible values for each parameter (especially the parameters settling resistance R and v_melt (by the way, what is the unit of v_melt as it is denoted as a speed?)). Are the bounds based on literature values, experience, etc.?

The chosen upper and lower parameter bounds are based on literature (e.g. for new snow density) or based on our previous experience with the model for the model specific parameters such as the settling resistance. For example, looking at the layer boundaries for different values of R gave us a qualitative feel for the range of reasonable values of this parameter. We now state this in the manuscript (Lines 158-160).

In an empirical model it is probable that some model parameters have no reference in reality. The same is true for the parameter 'v_melt' in SWE2HS. Therefore, we think it is not meaningful to attribute a unit to 'v_melt'.

- 163: The amount of your parameter set is high and sufficient. However, how did you come up with exactly 114688 parameter sets? Please introduce shortly the method of Saltelli (2002).

We expanded the description of the sensitivity analysis methodology and now briefly describe the method of Saltelli (2002) which is an extension of the low-discrepancy quasi-random Sobol' sequence and generates uniformly distributed samples of the parameter hypercube space (Lines 184-187).

For reduced error rates in the sensitivity index calculation, the number of samples N ideally is a power of base 2. For our experiment, we chose $N = 2^{13} = 8192$. The resulting length of the series is given by $L = N * (2*D + 2)$ where D is the number of parameters in the parameter space. This leads to the number of L=114688 parameter sets.

- 197ff: How often did you have the issue of filling up the data with linear interpolation (<= 5 days) and how often did you have data gaps?

See the table below for the number of gaps in HS and SWE in the automatic stations data set:

| Gap length | Number of gaps in HS | Number of gaps in SWE |
|:---:|:---:|:---:|
| 1 | 24 | 39 |
| 2 | 13 | 17 |
| 3 | 4 | 8 |
| 4 | 3 | 8 |
| 5 | 0 | 5 |
| 9 | 1 | - |
| 11 | 1 | - |
| Sum: | 39 | 77 |

We briefly mention the numbers of the interpolated SWE gaps in the manuscript (Line 235f).

- Figure 2: Which additional information can we get with the insets – are they really necessary?

Since the scatter plots are prone to overplotting problems, we decided to show the point densities in the insets. Showing only the densities was unsatisfactory because no outliers could be detected. Overlaying the densities, scatter plots, and linear fit in one plot was too cluttered, so we chose the inset layout as a compromise and would like to stick with it.

- 233f: Please insert a reference on the Akaike Information Criterion.

We inserted references to Akaike (1998) and Cavanaugh & Neath (2019) (Line 282).

- Figure 3: Please mention in the text the low $R^2$ of <0.4 for the validation data set for the month October. What could be the reason for the low $R^2$?

We now mention the low $R^2$ score for the month of October for the validation data set (Line 260) and added a short discussion of the potential reasons to Section 5.2 (Lines 331 - 335). One reason for the low $R^2$ score in October is probably the increased importance of new snow density. The snowfall density can have a high variability. This in completely unpredictable when keeping new snow density fix or independent of temperature or similar as done in the SWE2HS model. In this data set, October snow

cover is often characterized by ephemeral snowfall events, where new snow density is more important due to the lack of settled older snow layers.

- Section Conclusion: This part is rather short. As you give a short summery, please add also a sentence on the calibration part.

We added a few sentences on the calibration part to the conclusion (Lines 390-393)

**Replies to comments from Anonymous Reviewer #2**

**Summary**

The manuscript entitled '*An empirical model to calculate snow depth from daily snow water equivalent: SWE2HS 1.0*' presents an empirical model to derive daily snow depth (HS) from daily snow water equivalent (SWE) only. Density of each layer is modelled via an exponential function. Additionally, changes to due to overburden stress and SWE losses from runoff are considered. The model is calibrated using data from Switzerland and validated using a different dataset from automated weather stations in the European Alps (Austria, France, Germany, Switzerland). RMSE, R2 and Bias of the modeled HS with the obtained optimized parameters against the validation dataset are 20.5cm, 0.92 and 2.5cm compared to only 8.4cm, 0.97 and 0.2cm against the calibration dataset. The manuscript is well prepared but is lacking some details which should be addressed prior to being considered for publication.

**General comments**

- The model relies on HS records that are converted to SWE following Winkler et al. 2021 but this model is not described in sufficient detail to understand its impact (or lack thereof) on the optimization of the SWE2HS model. Please add short description of key/pertinent elements of the ΔSNOW model.

We added a short description of the ΔSNOW model by Winkler et al. (2021) to Section 3.3.1 (Lines 196 - 204).

- The paper converts daily HS records to SWE following Winkler et al. 2021 and uses these modeled SWE data, corrected using biweekly manual SWE measurements, to calibrate their SWE2HS model. RMSE and bias between modeled and measured SWE is 30.0mm and -1.09 mm. The model is well tuned to the calibration dataset but has larger errors compared to the validation data. Could some of this be due to the modeled SWE? The current text gives the impression that such differences are mainly due to the in situ SWE and not to the model which I find to be a bit simplistic especially given the relative magnitudes of the above errors relative to the SWE2HS calibration and validation statistics. The impact of the differences in modeled vs observed SWE in model calibration and thus accuracy should be discussed as a limitation in Sect 5.2.

We added a paragraph to Sect. 5.2 where we discuss the implication of calibrating the model on model output which is also associated with errors (Lines 317-322).

- The model was trained on the Switzerland data and then validated using an entirely different dataset covering a different spatial domain which adds additional complexity when interpreting the results. It's not clear whether (or to what extent) the validation results are due to regional variability, modeled SWE for calibration vs measured SWE for validation, etc. Figure 4 shows clear differences between the

calibration and validation datasets. Did the authors consider dividing the Switzerland data into calibration and validation datasets (and also perhaps divide the validation dataset into calibration and validation) to untangle some of these issues.

We have included a second validation data set which is consisting of the additional years from the Swiss manual observer station records longer than 15 years. This allows to asses the calibration performance more easily and in more depth. The Figures 4, 5, 6 and Table 3 now show results from all three data sets and we adapted the text in the results and discussion in order to reflect the third data set.

On the second validation data set, the model performs as good as on the calibration data set which indicates, that regional variability probably has a large contribution to the decreased performance on the AWS validation data set. We also state this in the discussion (Lines 315 - 317).

• Results could be discussed in greater detail, perhaps with additional sites considered (see specific comment regarding figures). The authors point out in which months the model accurately does or does not describe HS but offers little explanation as to why. The '*why*' would help users understand the strengths and limitations of the model.

We added several parts to the discussion where we pick up general and specific points from both reviewers regarding the 'why'. Additionally, we show more layer plots of station years from all three data sets (Figures 3, A1, and A2). We hope these changes help readers and potential users of SWE2HS to get a better understanding of the strengths and weaknesses of the model.

**Specific comments**

Figures:

• Sect 3.1 and 3.2 -> it would be helpful to provide a map showing the locations of the calibration and validation sites that were used in the analysis.

We added the following map which shows the stations that were used for the calibration data set and for the two validation data sets (new Figure 2 in manuscript)

[Figure]

- Figure 1: It might be instructive to present both a '*typical*' site as well as an '*excellent*' and '*poor*' site. This might help illustrate a wider range of conditions where the model performs well and poorly. Could be added in the main text or as a supplementary figure.

We have added five more station-years from the automatic stations calibration data to the figure, showing shallower and deeper snowpacks and cases where the model did not perform as well as for the Kühroint station in the winter of 2020/21. Additionally, we have added two supplementary figures showing 6 example winters from the calibration data and manual stations validation data set, respectively (Figures A1 and A2).

Abstract:

- Please add some specifics about the SWE2HS model. i.e. that it relies only on SWE, uses exponential settling functions and considers changes due to overburden stress and SWE losses from runoff.

This was totally missing and we admit it is important to get the basics of the model just by reading the abstract. We added a short description of the basic principles to the abstract (Lines 7-9).

- Please note the regional applicability of the SWE2HS model.

We added the following sentence to the abstract: „Owing to its empirical nature, SWE2HS should be only used in regions with a similar snow climatology as the European Alps or has to be recalibrated for other snow climatological conditions." (Line 17f)

- L10: please note the locations and type of validation data (i.e. AWS Austria, France, Germany, Switzerland)

We mentioned AWS and European Alps already. However, we are more specific now and mention the countries in which the AWSs are located in L. 11.

- The authors present the need for this model in terms of tourism applications [L25]. Please mention why SWE (and not HS) is often desired and is the focus of many of the models mentioned. i.e. climate, hydrology, HS easier to measure operationally, etc.

We added a sentence which makes clear that usually SWE is the variable of interest for climatological analyses and that snow depth is often measured more easily (Lines 33f and 36f). However, when snow depth is needed on a spatially gridded domain, measurements are usually not available.

- I found it a bit difficult going back and forth between calibration, calibration data and then validation data and methods. Perhaps group methods and data together? Or add a sentence or two to the first part of Sect 3 that touches on validation (in addition to calibration). See also comment about additional Figure for the cal/val data.

We have added additional subsection headings to Section 3 to improve the logic of the structure. We also mention in the introduction of Section 3 the validation part. We hope that these changes, together with the map of station locations, will improve the clarity for the reader.

**Specific editorial suggestions**

- L42: suggest 'The first category is purely empirical whereby densification dynamics are described via exponential settling functions.'

Changed accordingly (L. 52).

- L77: suggest 'The maximum density starts with an…'

Changed accordingly (L .92).

- L78: suggest 'The maximum density starts with an initial value at creation time of the layer and subsequently increases towards…'

Changed accordingly (L. 92f).

- L80: Implications of this simplification? i.e. removing the top layer instead of decreasing all layers.

Thank you for bringing up this point. Please see our answer to the comment on L.80 from Reviewer 1.

- L85: suggest 'The density of a layer at day *i* asymptomatically converges…'

Changed accordingly (L. 99).

- L90: suggest 'The maximum density to which the density of a snow layer converges, $\rho_{max}$ in Eq. 1, also evolves over time.'

Changed accordingly (L. 104).

- L91: suggest 'is that snow which has experienced a high load reaches a higher…'

Changed accordingly (L. 105)

- L93: delete one of the 'the' in 'The third assumption is that the…'

Done.

- L96: 'Afterwards, $\rho_{max}$ increases towards …'

Changed accordingly (L. 110).

- L104: 'Whenever SWE in the snowpack decreases we assume that the snowpack wet entirely and we attribute…'

Changed to "Whenever SWE in the snowpack decreases, we assume that the entire snowpack is wet since we attribute all SWE losses to runoff." (L. 118).

- L168: 'To calibrate the SWE2HS…'

Changed accordingly (L. 193).

- L192: 'As a validation dataset' or 'For validation…'

Changed to "As a second validation data set" (L. 229).

**Revisions not related to reviewer comments**

- Upon the impulse from this manuscript, the Austrian Research Centre for Forests BFW in Innsbruck has decided to publish the data from the Wattener Lizum field site (Hagen et al. 2023). Prior to publication, the BFW quality checked and corrected the data. Due to slightly different quality check and correction methods, this newly published data set is not identical to the one used in the discussion paper manuscript. Accordingly, we changed the Wattener Lizum data in the AWS validation data set to be consistent with the one published in Hagen et al. (2023). With the new data, the number of used years from the Wattener Lizum station increased from 8 to 12 (see Table 1 in the Manuscript). However, the score metrics on the validation data set did not change considerably compared to the version with the previously used Wattener Lizum data.

- We published the to date not publicly available data from the Bavarian Avalanche Service stations and the manual stations calibration and validation data used for this paper on envidat.ch. The data is accessible through https://doi.org/10.16904/envidat.394 (Aschauer & Marty 2023).

- We changed the layout of Figure 7 (Total sensitivity indices results) in order to save space and to make the sensitivity indices better comparable between R^2 and BIAS.

**References**

Akaike, H. (1998). Information theory and an extension of the maximum likelihood principle. Selected papers of hirotugu akaike, 199-213.

Aschauer, J. and Marty, C.: SWE2HS model calibration and validation data, https://doi.org/10.16904/envidat.394, 2023.

Hagen, K., Köhler, A., Fromm, R., and Markart, G.: Daily snow water equivalent and snow depth data from the valley Wattental in the Tuxer Alpen, Tyrol, Austria [dataset], Zenodo, https://doi.org/10.5281/zenodo.7845618, 2023.

Cavanaugh, J. E., & Neath, A. A. (2019). The Akaike information  criterion: Background, derivation, properties, application,  interpretation, and refinements. Wiley Interdisciplinary Reviews: Computational Statistics, 11(3), e1460.

Hock, R. (2003). Temperature index melt modelling in mountain areas. Journal of hydrology, 282(1-4), 104-115.

Michel, A., Aschauer,  J., Jonas, T., Gubler, S., Kotlarski, S., and Marty, C. (2023). SnowQM 1.0: A fast R Package for bias-correcting spatial fields of snow water equivalent using quantile mapping. Submitted to Geoscientific Model Development Discussions.

Steinkogler, W., Fierz, C., Lehning, M., & Obleitner, F. (2009). Systematic assessment of new snow settlement in SNOWPACK. Proceedings of International Snow Science Workshop, Davos 2009, 132-135.